# Mow the Grass at the Mouse’s Peril: Diversity of Small Mammals in Commercial Fruit Farms

**DOI:** 10.3390/ani9060334

**Published:** 2019-06-08

**Authors:** Linas Balčiauskas, Laima Balčiauskienė, Vitalijus Stirkė

**Affiliations:** Nature Research Centre, Akademijos 2, LT 08412 Vilnius, Lithuania; laima.balciauskiene@gamtc.lt (L.B.); vitalijus.stirke@gamtc.lt (V.S.)

**Keywords:** small mammals, commercial fruit farms, diversity, dominance, relative abundance, agricultural practices

## Abstract

**Simple Summary:**

For the first time in the Baltic countries, small mammal communities were evaluated in commercial orchards and berry plantations to test for the influence of crop type and intensity of agricultural practices. Out of ten species registered in the fruit farms, the most dominant were common vole and striped field mouse, confirming the spread of the latter species in the country. Small mammal diversity and abundance were not dependent on crop type but decreased in line with the intensity of agricultural practices, not being found in the most intensively cultivated farms. Unexpectedly, small mammal diversity in apple orchards exceeded the diversities found in most types of forests and was significantly higher than in crop fields.

**Abstract:**

Small mammals are not only pests but also an important part of agricultural ecosystems. The common vole is a reference species for risk assessment of plant protection products in the European Union, but no data about the suitability of the species in the Baltic countries are present so far. Using the snap-trap line method, we evaluated species composition, abundance, and diversity of small mammal communities in commercial orchards and berry plantations in Lithuania, testing the predictions that (i) compared with other habitats, small mammal diversity in fruit farms is low, and (ii) the common vole is the dominant species. The diversity of small mammals was compared with control habitats and the results of investigations in other habitats. Out of ten small mammal species registered, the most dominant were common vole and striped field mouse. Small mammal diversity and abundance increased in autumn and decreased in line with the intensity of agricultural practices but were not dependent on crop type. In the most intensively cultivated fruit farms, small mammals were not found. The diversity of small mammal communities in fruit farms was significantly higher than in crop fields and exceeded the diversities found in most types of forests except those in rapid succession.

## 1. Introduction

In 2017, there were 25,980 ha of commercial fruit farms in Lithuania (including 9851 ha of berry plantations), which yielded 99,215 tons of production [1]. However, despite commercial fruit farms being an important agricultural industry, not all aspects of their ecology have been subjected to scientific research in Lithuania. We present the first results of investigations into small mammals in commercial orchards and berry plantations in the Baltic countries.

Being a part of the food web and providing ecosystem functions in agricultural ecosystems [2], small mammals also are pests of many crops [3,4]. Their negative impact is not limited to crop damage [5] but also includes their role in the dispersal of weed seeds and being a reservoir for various pathogens [6]. Monetary losses from the detrimental activities of rodents on crops may reach billions of US dollars worldwide [7].

Despite a decline of farmland biodiversity in Europe [8], the diversity of small mammal species in the agricultural landscape may be considerable [3,9,10,11,12]. Within this diversity, however, only several species of *Apodemus* mice and voles of the genus *Microtus*, mainly common voles (*Microtus arvalis*), are recognized as major pests [5,13,14,15,16].

In recent decades, a number of changes in European agricultural landscapes have influenced small mammal communities. There are tendencies that enhance land capacity for small mammals, such as the Entry Level Scheme Tier of Environmental Stewardship, whereby 1% of cropland is converted to wildlife habitats [8] or other habitats with various levels of success [17]. Management measures of small mammal communities in the agro-landscape should be related to knowledge of the factors that influence their formation and survival [18]. This is important not only from the point of view of nature protection but also for pest control [19,20].

In the Czech Republic, the highest abundance and diversity of small mammals was observed in fallow lands and crops with long-term vegetation [11]. In Romania, vegetation traits were the most important factors influencing small mammal communities [18]. In Mexico, orchards exceeded crop fields and grasslands in terms of small mammal diversity, species richness, and abundance [21]. In Norway, the highest number of small mammal species was unexpectedly found in cultivated fields [22]. 

In general terms, small mammal diversity in agro-landscapes depends on several factors: (a) disturbance of the habitat [23], with medium levels of disturbance possibly supporting higher small mammal diversity [24]; (b) application of rodenticides and mechanical measures such as mowing, soil scarification, and traps, though these had no permanent effect on voles [15,25,26,27]; (c) nonlethal measures that affect the behavior and reproductive success of animals [28], such as biological pest control or the provision of owl boxes [29]; (d) time required for communities to restore from surviving individuals or migration from neighboring habitats [30,31,32]; (e) crop type [11]; and (f) biology and ecology of small mammal species [30].

Following [13], we presume that small mammals are able to colonize agricultural habitats. This is especially important for small, prolific, resilient to disturbance, and cyclic species of voles [27]. Their proportion in small mammal communities in orchards may be diminished by using living mulch [16], the use of repellents on the tree trunks, and the removal of grass around the bases of the trees [33]. The use of barriers, traps, and rodenticides should equally affect all small mammal species [27]. However, the side effects of rodenticides are well known and debated [7,34,35], so the use of repellents or attractants to encourage the exit of small mammals from the territory seems a more positive solution [4]. The effects of habitat manipulation, supplemental feeding, and natural predation on small mammal diversity are not well known, as these means were targeted at reducing the abundance of small mammals [27].

Compensation in small mammal communities ensures species replacement after various disturbances [12]. Thus, migration from surrounding territories is an important factor that determines diversity. Edge habitats and hedgerows enhance diversity in agricultural lands [36], as does the creation of field margins; these substantially increase the species richness and abundance of voles, mice, and shrews [8,31]. Fallow habitats support abundant, diverse, and stable small mammal communities [11] but do not significantly alter species composition [18]. However, rodent activity peaks are characteristic also “in the middle of the orchard regardless of field border type” [31]. Such patterns emphasize the relevance of not only changes in the amount of agricultural land (decreasing in Central Europe, see [11]) but also its fragmentation and patchiness [21].

Our aim was to assess the species composition, abundance, and diversity of small mammal communities in orchards and berry plantations in Lithuania, this representing other Baltic countries with similar climate and agricultural traditions. Based on literature sources, we predicted that (i) there would be low small mammal diversity in commercial gardens and berry fields compared with other habitats and (ii) common vole should be the dominant species. Comparing the summer and autumn seasons, we evaluated if the species composition, abundance, and diversity of small mammal communities are influenced by crop type and the intensity of agricultural practices used.

## 2. Material and Methods

We conducted our study in commercial orchards and berry plantations in Lithuania at the beginning of summer (6–22 June) and in autumn (4–20 September and 1–11 October) in 2018. Comparing with control habitats located next to the fruit farms, we surveyed small mammal species composition, diversity, and relative abundance in apple and plum orchards, as well as in currant, raspberry, and highbush blueberry plantations.

### 2.1. Study Sites

Fifteen study sites with 16 trapping locations were selected across Lithuania (Figure 1), these representing central (55.58° N, 23.86° E), northern (55.97° N, 25.01° E), eastern (55.60° N, 25.27° E), southern (54.29° N, 24.24° E), and western (55.44° N, 22.22° E) parts of the country, as well as a variety of culture types and agricultural practices, such as grass mulching, mowing, soil scarification, and application of plant protection agents and rodenticides (Table 1, Appendix A
Figure A1, Figure A2, Figure A3 and Figure A4). Rodenticides were used by farmers at six of the study sites. At site 13, in addition to very intensive treatment, including soil scarification of interlines, farmers provided perches for birds of prey along the perimeter fencing and recordings of the voices of birds of prey were played regularly.

Control habitats were meadows, with grass mowing as the only factor possibly influencing small mammal communities at ten sites.

We evaluated the intensity of agricultural practices as follows: high intensity (frequent application of two or more of the above-listed measures, including rodenticides), medium intensity (two listed measures during the crop season, once or several times), and nonintensive (removal of grass only).

### 2.2. Small Mammal Trapping

Small mammals were trapped using 7 × 14 cm wooden snap traps set in lines of 25 traps, each set 5 m apart as in [37]. The number of lines per site depended on the size of the fruit farm, ranging from one to four. Despite the warning of [38,39] about the effects of removal trapping of small mammals on abundance and diversity, our data show no influence of short trap lines on the diversity and abundance of small mammals, even after long-term trapping of up to three times per year [40,41,42,43]. In this study, we trapped two times. Several sites had no small mammals in the first trapping session; thus, the first trapping could not have negatively influenced the second trapping session. Traps were baited with bread crust with raw sunflower oil, set for three days, and checked once per day. According to [37], the bait should be changed after rain or when consumed by other mammals, birds, insects, or slugs. The total trapping effort was 8880 trap days (Table 2). The relative abundance of small mammals was expressed as standard capture rates to number of animals/100 trap days of the first day of trapping. As there was no rain on the first day of trapping, correction of abundance values for sprung traps was not used; at sites 13 and 15, no small mammals were trapped in summer and autumn.

Standard measures (body mass, body length, tail length, hind foot length, and ear length) were taken before dissection. Species were identified morphologically, with specimens of *Microtus* voles identified by their teeth. Juveniles, subadults, and adults were identified under dissection based on body weight, the status of sex organs, and atrophy of the thymus, the latter of which decreases with animal age [44].

We used snap trapping to obtain material for the analysis of chemical elements in the bodies of the small mammals and to search for various pathogens (data not used in the current publication). The study was conducted in accordance with the principles of Lithuanian legislation for animal welfare and wildlife. Permission to trap wild small mammals was provided according to Regulation No. 6 (2018-02-02) of the Ministry of the Environment of the Republic of Lithuania.

### 2.3. Data Analysis

Species richness was expressed as the number of trapped species, while the Shannon–Wiener diversity index, H, on the base of log_2_ [45] was used as a measure of the diversity of the small mammal community and the dominance index, D, as a measure of dominance [19,45,46]. The diversity of the community was compared to other habitats and territories of varying sizes in Lithuania ([40,41,42,43,47,48]; Appendix B
Table A1). Diversity estimates (including bootstrap, yielding 95% confidence interval) were calculated using PAST ver. 2.17c (Ø. Hammer, D.A.T. Harper, Oslo, Norway) [46,49] and EstimateS ver. 9.1.0 (R.K. Colwell, Connecticut, CT, USA) [50,51]. Rarefaction was used according to individual-based data to produce species accumulation curves and compare species richness between fruit farms, berry fields, and control habitats and between fruit farms with different intensities of treatment. Rarefaction eliminates the influence of the sample size on the diversity estimate (number of individuals or trapping effort, thus the effect of the farm size at the same time). Differences of community composition were evaluated using chi-square statistics in PAST.

We applied factorial ANOVA to relative small mammal abundance, diversity, and dominance as dependent variables across the research sites to test the possible cumulative influence of categorical predictors—season, crop type, and intensity of agricultural practices. Hotelling’s T^2^ was used for multivariate testing. Thereafter, significant factors were used for ANOVA analysis, and Tukey’s HSD with unequal N for post hoc comparisons [52]. The confidence level was set as *p* < 0.05. Calculations were done in Statistica for Windows, ver. 6.0 (StatSoft, Inc., Tulsa, OK, USA).

## 3. Results

We trapped 512 individuals of 10 small mammal species (Table 3). Overall, orchards and plantations were inhabited by all recorded species but were dominated by common vole (*M. arvalis*) (31.0%), striped field mouse (*Apodemus agrarius*) (27.1%), and yellow-necked mouse (*Apodemus flavicollis*) (19.5%). Control habitats were characterized by nine species, with pygmy shrew (*Sorex minutus*) not trapped, and these were dominated by *A. agrarius* (36.8%), *M. arvalis* (18.7%), *A. flavicollis*, and bank vole (*Myodes glareolus*) (both 13.9%). Species composition significantly differed between the fruit farms and controls (χ^2^ = 25.7, df = 9, *p* < 0.01), the main difference being higher numbers of *M. arvalis* in the former and higher numbers *A. agrarius* in the latter habitats. The diversity of small mammals in orchards and plantations was the same as in control habitats, with dominance also not differing (Figure A5A). No significant differences in diversity indices between the control habitats and commercial fruit farms were found in the separate seasons of summer (Figure A5B) and autumn (Figure A5C). In autumn, however, species accumulation curves showed that a higher diversity was reached in control habitats with a smaller number of trapped specimens.

Testing the cumulative influence of season, crop type, and intensity of agricultural practices on relative small mammal abundance, diversity, and dominance showed a significant dependence on season (Hotelling’s T^2^ = 0.353, df = 3, *p* = 0.001) and intensity (T^2^ = 0.316, df = 6, *p* = 0.02) but not habitat type (T^2^ = 0.207, df = 12, *p* = 0.59). Univariate tests showed a significant influence of season (ANOVA, F = 17.01, df = 1, *p* < 0.001) and intensity of agricultural practices (F = 4.09, df = 2, *p* < 0.05) on the relative abundance of small mammals, as well as the influence of season (F = 9.65, df = 1, *p* < 0.01) on diversity and the influence of intensity of agricultural practices (F = 3.78, df = 2, *p* < 0.05) on dominance in small mammal communities.

### 3.1. Small Mammal Diversity and Abundance in Relation to Crop Type

Small mammal species richness was highest in the apple orchards and their control habitats, then declined in the following order: raspberry plantations and their controls, currant plantations, currant controls, and plum orchards and their control habitats. No small mammals were trapped at all in the highbush blueberry plantation (Table 3). Thus, our first prediction was not confirmed.

Small mammal diversity (Table 3) in the apple orchards was higher than in plum orchards, currant plantations, and raspberry plantations. Plum orchards had the same small mammal diversity as currant and raspberry plantations, while diversity in raspberry plantations exceeded that in currant plantations. Species accumulation curves (Figure 2A) fully confirmed the diversity differences. Differences in the small mammal diversity between crops and respective control habitats were not found.

Dominance in the small mammal community was most notably expressed in the currant plantations (Table 3), where *M. arvalis* constituted nearly 70% of all trapped individuals (64% in the control habitats). This species also dominated in plum orchards (46%), whereas *A. agrarius* dominated in raspberry plantations (48%). Control habitats of the raspberry plantations were dominated by *A. flavicollis* (33%), *A. agrarius* (27%), and *M. glareolus* (24%). Four small mammal species were represented almost equally in apple orchards (Figure 2B). Control habitats of the apple and plum orchards were dominated by *A. agrarius* (over 40% of all individuals in both controls). Other small mammal species were not trapped in large numbers. Thus, our second prediction about the dominance of *M. arvalis* was confirmed but not for all habitats.

The relative abundance of small mammals varied across the study sites and ranged between 0 and 12 individuals per 100 traps/day in control habitats and commercial fruit farms in summer. In June, no small mammals were trapped in fruit farms at seven sites (Nos. 1, 2, 5, 11, and 13–15, representing all crops). In the autumn, the relative abundances were 0–36 in commercial fruit farms and 0–30.7 individuals per 100 traps/day in control habitats. Fruit farms at three sites (Nos. 5, 10, and 13) were not inhabited by small mammals in the autumn, two of which were unoccupied in both seasons. The averages of the relative abundance of small mammals between the respective controls and fruit farms in summer and autumn did not differ (Table 3).

### 3.2. Small Mammal Diversity and Abundance in Relation to Intensity of Agricultural Practices

The number of small mammal species, recorded in the sites with medium and high intensity of agricultural practices, was lower than in respective control habitats. The small mammal diversity in the fruit farms with high-intensity practices was the lowest (Table 4), differing from the respective control habitats and fruit farms with medium- and low-intensity practices. Low- and medium-intensity practices had similar effects on small mammal diversity. In fruit farms with medium intensity of agricultural practices, it did not differ from control habitats. Species accumulation curves (Figure 3A) confirmed that the highest diversity of small mammals was in control habitats and habitats with medium intensity of disturbance.

The dominance index was highest in small mammal communities living under the highest pressure of agricultural practices (Table 4). In these fruit farms, three small mammal species—*M. arvalis*, *A. flavicollis*, and *A. agrarius*—constituted over 95% of all trapped individuals, while respective control habitats were dominated by *A. agrarius*, representing 47% of all trapped individuals (Figure 3B). Over 85% of all trapped individuals in fruit farms with low pressure were represented by *A. agrarius*, *M. arvalis*, and *M. glareolus*, with the same species dominating respective control habitats. Habitats with medium pressure of agricultural activities were dominated by *M. arvalis*, and their respective control habitats by *A. agrarius* (Table 4).

The average relative abundances of small mammals in summer and autumn were highest in fruit farms with low intensities of agricultural practices (Table 4), exceeding those in sites with medium intensity by about 4 times and those in sites with high disturbance by nearly 10 times in summer and 2–4 times in autumn. At the beginning of summer, no small mammals were trapped in four fruit farms with high intensity (Nos. 2, 11, 13, and 15) and three fruit farms with medium intensity (Nos. 1, 5, and 14) of agricultural practices. In autumn, the number of farms with no small mammals was two (Nos. 10 and 13) and one (No. 5), respectively. Small mammals were always present at all sites with a low intensity of measures. Thus, intensity had a negative effect on small mammal presence.

The intensities of agricultural practices were also well reflected by the variation of relative abundance of small mammals across sites. In summer, relative abundances were 0.3–12 ind. in fruit farms with a low intensity of influence, 0–4 ind. in medium intensity, and only 0–1.1 ind. in the highly affected sites. In the autumn, the respective figures were 4–36, 0–9, and 0–28 ind. The averages of the relative abundance of small mammals between the respective controls and farms with different intensity of agricultural practices in summer and autumn did not differ (Table 4).

## 4. Discussion

As recognized in the European Common Agricultural Policy Reform [53], one of the dominant threats to biological diversity is intensive agriculture, resulting in a decline of small mammal populations in Europe [54]. Thus, even while recognizing negative small mammal impacts such as crop damage and monetary losses [3,7], the current position is to maintain their communities, even in agricultural ecosystems, as providers of ecosystem services [2]. Therefore, chemical-based pest management methods are being changed to those based on small mammal ecology [55] or biological control [29].

Knowledge regarding the species and their habitats should assist in the development of new rodent pest management strategies [56]. In disturbed habitats with varying agricultural practices, species ecology will be different [53]. Knowledge of the driving factors in small mammal ecology would allow harmonization of conservation needs and economic necessities [18].

We found that small mammal diversity and relative abundance were mainly defined by season and intensity of agricultural practices but not by crop type. In our study, soil scarification of interlines was used in 25% of investigated sites, while rodenticides in approximately 40%, plant protection agents in nearly 75%, and grass mowing at all sites. In the current analysis, usage of rodenticides was treated as any other agricultural activity and not analyzed separately. The application of rodenticides, however, had only a limited effect, as in the sites where rodenticides were used, three species of small mammals were recorded in spring and six species in autumn.

The main agricultural activity in our study was grass mowing. Depending on whether the mowed grass was removed from the fruit farm, used as mulch, or left in place, the impact of mowing on small mammals was different. Small mammal numbers decline rapidly after mowing, but up to 27% of small mammals remain if the cut grass is not moved [57]. Habitats that are not mown every year and ecological compensation areas in agricultural landscape support the highest numbers of small mammals [54]. Rodents respond to decreased vegetation height by reducing their activity and changing their feeding pattern [15]. Unmowed grasslands are characterized by lower giving-up densities and differences in the use of territory; they are less used at night, possibly accounting for owl activity [58]. Using the analogy of understory, we may expect changes in reproduction parameters [59]. Finally, decreased mowing intensity after the crop season in commercial fruit farms may facilitate easier recolonization of the territory from surrounding habitats. In our study, the example is the migration of *A. flavicollis* into the raspberry plantation at site 11 from the adjacent forest in autumn.

In Lithuania, two dominant species, *M. arvalis* and *A. agrarius*, were found in most of the investigated commercial fruit farms. The exception was the old apple orchard at site 7 (Figure A3d) with high grass cover and almost no agricultural activities, which was dominated by *M. glareolus*. *M. arvalis* is one of the most numerous species in Central Europe [11], which may consume nearly 50% of the annual production of alfalfa during outbreaks [60]. An expansion of *A. agrarius* has been recorded in other European countries [61], and this requires special attention in Lithuania [43].

Based on our long-term small mammal studies [40,41,42,43,44,47,48], we suppose that the number of trapping sites (spatial replications) was enough to represent possible temporal changes in small mammal diversity and abundance (i.e., the results are representative). Earlier, different small mammal densities and number of species were found within the same year in forests, swamps, or meadows located several kilometers from each other [41,42,47,48].

In commercial fruit farms, our results indicate a higher species richness and an equal diversity and dominance in small mammal communities when compared with control habitats but with different species composition and relative abundance.

We compared our results with other agricultural and natural habitats across the country using published data from 125 trapping sessions (Appendix B
Table A1), which represent long-term trapping [41], forest succession [42], flooded and nonflooded habitats [44], various habitats in protected territories [47], and data from small mammal monitoring in the agro-landscape [48].

We found that the small mammal diversity in our control meadows was in line with that in natural, seminatural, and sown meadows evaluated in the long-term investigations, lower than in shrubby or reforesting meadows but much higher than in meadows of the agro-landscape (all mentioned differences significant with *p* < 0.001). The small mammal diversity that we found in apple orchards was significantly higher than that recorded in various forest types, with the exception of young forest growing in former meadows and regrowing after forest clear–cutting, where the succession between ecosystem stages shapes the small mammal community [42,47]. Small mammal diversity in the plum orchards was in line with most forest types, while small mammal diversity in the berry plantations was significantly lower than in meadows and in line or exceeding those in cropland, forests, and other habitats.

## 5. Conclusions

Out of ten small mammal species registered in commercial fruit farms in Lithuania, the most dominant were *M. arvalis* and *A. agrarius*. The diversity of small mammal communities in the farms was higher than in crop fields, while the diversity in the fruit orchards exceeded or was in line with that of the forests. The diversity in berry plantations was in line or exceeded cropland and forests but was lower than in meadows. Both diversity and abundance of small mammals were dependent on the season and the intensity of agricultural practices, but neither were dependent on crop type. In the most intensively cultivated fruit farms, small mammals were not found.

## Figures and Tables

**Figure 1 animals-09-00334-f001:**
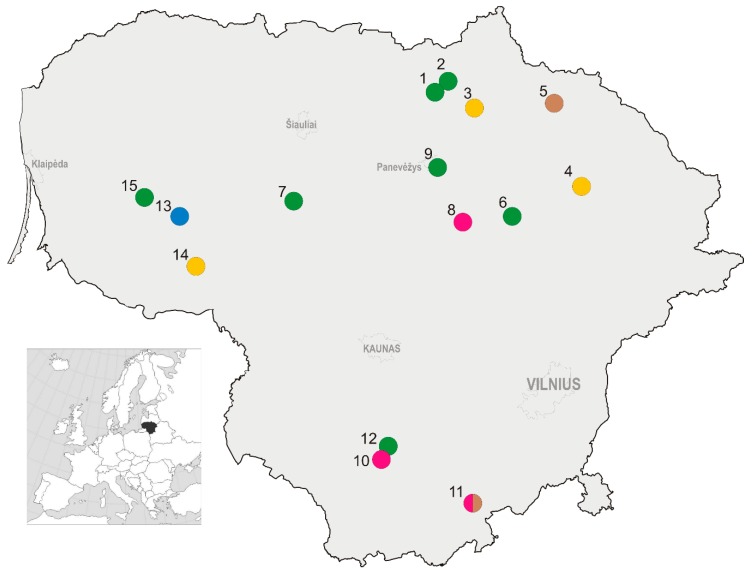
Location of the study sites in Lithuania: 1—Aukštikalniai, 2—Pajiešmeniai, 3—Gaižiūnai, 4—Užpaliai, 5—Kalpokai, 6—Šeimyniškiai, 7—Šedbarai, 8—Šiekštinės, 9—Dembava, 10—Užubaliai, 11—Barčiai, 12—Luksnėnai, 13—Naujasis Obelynas, 14—Gaurė, 15—Kvėdarna. Crop type is indicated by color: green—apple orchards, brown—plum orchards, magenta—raspberry plantations, yellow—currant plantations, and blue—highbush blueberry plantations. Location coordinates are not presented to ensure privacy of the owners.

**Figure 2 animals-09-00334-f002:**
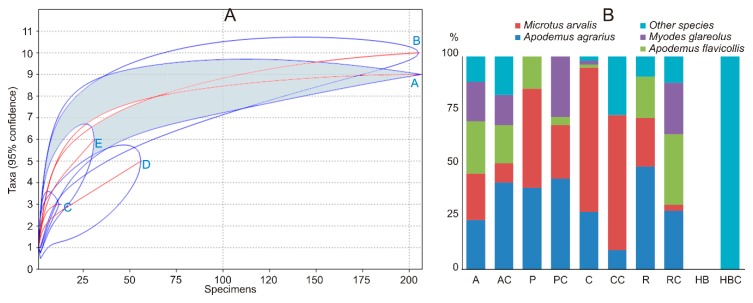
Small mammal species accumulation curves (**A**): A—control, B—apple orchards, C—plum orchards, D—currant plantations, E—raspberry plantations, and share of the dominant species in the control habitats and commercial fruit farms in Lithuania, 2018 (**B**): A—apple orchards, AC—apple orchard controls, P—plum orchards, PC—plum orchard controls, C—currant plantations, CC—currant plantation controls, R—raspberry plantations, RC—raspberry plantation controls, HB—highbush blueberry plantations, HBC—highbush blueberry plantation controls.

**Figure 3 animals-09-00334-f003:**
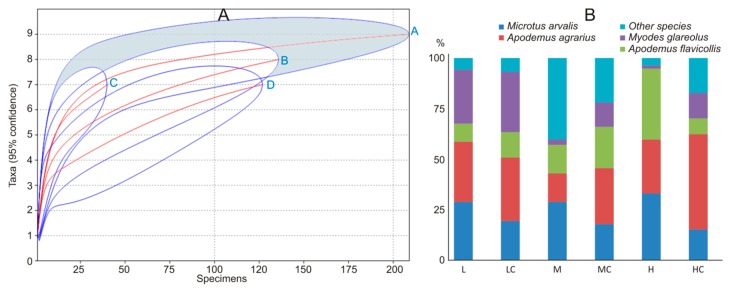
Small mammal species accumulation curves (**A**): A—control, B—low, C—medium, D—high intensity, and share of the dominant species in the control habitats and commercial fruit farms according to the intensity of agricultural practices (**B**): L—low intensity, LC—low-intensity controls, M—medium intensity, MC—medium-intensity controls, H—high intensity, HC—high-intensity controls.

**Table 1 animals-09-00334-t001:** Characteristics of study sites. Site numbers as in Figure 1.

Site No.	Crops	Age ^1^	Appendix Figure	Intensity ^2^	Agricultural Practices ^3^				Control Habitat ^4^	Appendix Figure	Mowing Practices ^5^	
					GM	S	PPA	RD			T	GR
1	Apple	O	A2a	M	+	−	+	+	MM	A4a	S	R
2	Apple	O		H	+	−	+	+	MM		LA	R/M
3	Currant	MD	A3a	L	+	−	−	−	MM	A4b	S	R
4	Currant	MD		L	+	−	−	−	MM		S	R
5	Plum	Y	A2d	M	+	−	+	−	NM	A4c		
6	Apple	O	A1a	H	+	−	+	+	MM	A4d	LA	R
7	Apple	O	A3d	L	+/−	−	−	−	NM			
8	Raspberry	MD	A3b	L	+	−	−	−	NM	A4e		
9	Apple	O	A2b	M	+	−	+	−	MM	A4f	S	R
10	Raspberry	Y	A1c	H	+	+	+	−	MM		LA	R
11	Raspberry	MD		H/M	+	+	+	−	FE			
	Plum	MD	A3c	L	+/−	−	−	−	MM		S	NR
12	Apple	O		H	+	−	+	+	MM		LA	R
13	Highbush blueberry	MD	A1b	H	+	+/−	+	+	MM		LA	R
14	Currant	MD	A2c	M	+	−	+	−	MM		S	R
15	Apple	MD	A1d	H	+	+	+	+	MM		LA	R/M

^1^ Age of the orchard: O—old, MD—medium, Y—young. ^2^ Intensity of agricultural practices on site: L—low, M—medium, H—high. ^3^ Measures used: GM—grass mowing, S—soil scarification of interlines, PPA—application of plant protection agents, RD—application of rodenticides. ^4^ MM—mowed meadow, NM—nonmowed meadow, FE—forest edge. ^5^ T—timing (LA—till late autumn, S—summer only), GR—grass removal (R—removed, NR—nonremoved, M—used as mulch). +: practice used. −: practice not used.

**Table 2 animals-09-00334-t002:** Summary of trapping results in 2018 in the commercial fruit farms and control habitats.

Season	Orchards and Plantations ^1^		Control Habitats		Total	
	TE	N	TE	N	TE	N
Summer	2970	46	1440	46	4410	92
Autumn	2925	257	1545	163	4470	420
Total	5895	303	2985	209	8880	512

^1^ TE—trapping effort, trap days, N—number of individuals trapped.

**Table 3 animals-09-00334-t003:** Small mammal diversity, numbers trapped, and relative abundance (recalculated to 100 traps per day) in commercial orchards, berry plantations, and control habitats.

Species	A ^1^	AC	P	PC	C	CC	R	RC	HB	HBC
*Sorex araneus*	3	6	0	0	1	0	0	3		2
*Sorex minutus*	2	2	0	0	0	0	0	0		
*Apodemus agrarius*	47	61	5	6	15	1	15	9		
*Apodemus flavicollis*	50	27	2	1	1	0	6	1		
*Micromys minutus*	1	1	0	0	0	0	1	1		
*Mus musculus*	0	1	0	0	0	1	1	0		
*Microtus agrestis*	8	10	0	0	0	0	0	0		
*Microtus arvalis*	44	13	6	7	38	7	7	11		
*Microtus oeconomus*	10	7	0	0	0	2	1	0		
*Myodes glareolus*	38	21	0	0	1	0	0	8		
Total, N ^2^	203	149	13	14	56	11	31	33		2
S	9	10	3	3	5	4	6	6	0	1
TD	3495	1665	300	225	1200	570	600	375	300	150
D (95% CI)	0.20 (0.18–0.23)	0.24 (0.18–0.24)	0.38 (0.35–0.62)	0.44 (0.35–0.59)	0.53 (0.41–0.64)	0.45 (0.32–0.83)	0.33 (0.20–0.38)	0.25 (0.20–0.37)		2
H (95% CI)	1.74 (1.64–1.86)	1.74 (1.62–1.86)	1.01 (0.54–1.07)	0.90 (0.60–1.08)	0.83 (0.64–1.17)	1.03 (0.30–1.24)	1.34 (1.17–1.76)	1.49 (1.19–1.76)		
RA_s_ ± SE	2.63 ± 1.65	4.95 ± 1.94	2.00	0.67 ± 0.67	0.56 ± 0.40	0	1.11 ± 0.80	6.00 ± 2.00	0	0
RA_a_ ± SE	11.03 ± 4.82	14.00 ± 4.07	6.67	17.33	6.67 ± 4.44	4.44 ± 2.47	12.44 ± 8.23	10.67 ± 5.05	0	2.67

^1^ Crops: A—apple orchards, AC—apple orchard controls, P—plum orchards, PC—plum orchard controls, C—currant plantations, CC—currant plantation controls, R—raspberry plantations, RC—raspberry plantation controls, HB—highbush blueberry plantations, HBC—highbush blueberry plantation controls. ^2^ N—number of individuals trapped, S—number of species, TD—trapping effort, trap days, D—dominance index, H—Shannon’s index of diversity, RA_s_—relative abundance in summer, RA_a_—in autumn. H differences: A–P (*t* = 5.66, *p* < 0.001), A–C (*t* = 7.24, *p* < 0.001), A–R (*t* = 2.91, *p* < 0.01), P–C and P–R (NS), C–R (*t* = 2.40, *p* < 0.05); A–AC (*t* = 0.05, NS), P–PC (*t* = 0.55, NS), C–CC (*t* = 0.34, NS), R–RC (*t* = 0.84, NS). RA differences in summer (ANOVA, F_5,25_ = 0.41) and autumn (F_5,25_ = 0.35) were not significant between habitats.

**Table 4 animals-09-00334-t004:** Small mammal diversity, numbers trapped, and relative abundance (recalculated to 100 traps per day) in fruit farms according to the intensity of agricultural practices. Control data presented in Table 3.

Species	Intensity of Agricultural Practices and Controls ^1^					
	L	LC	M	MC	H	HC
*S. araneus*	4	12		2		7
*S. minutus*	1	7	1	2		
*A. agrarius*	46		6	19	30	46
*A. flavicollis*	14	1	6	14	39	8
*M. minutus*		11			2	2
*M. musculus*		1			1	1
*M. agrestis*			8	8		1
*M. arvalis*	46	9	12	12	37	15
*M. oeconomus*	2	2	8	3	1	6
*M. glareolus*	36		1	8	2	12
Total, N ^2^	149	43	42	68	112	98
S	7	7	7	8	7	9
TD	1500	825	1650	870	2745	1290
D (95% CI)	0.26 (0.17–0.23)	0.22 (0.16–0.28)	0.20 (0.16–0.28)	0.18 (0.16–0.26)	0.30 (0.17–0.24)	0.27 (0.17–0.24)
H (95% CI)	1.48 (1.69–1.95)	1.65 (1.48–1.99)	1.72 (1.47–1.97)	1.84 (1.58–1.98)	1.31 (1.66–1.95)	1.64 (1.64–1.96)
RA_s_ ± SE	4.07 ± 2.08	4.00 ± 2.53	1.00 ± 1.00	3.17 ± 2.74	0.44 ± 0.22	2.56 ± 1.07
RA_a_ ± SE	15.60 ± 5.46	9.00 ± 7.23	2.58 ± 2.14	11.67 ± 4.01	8.51 ± 2.71	11.52 ± 2.86

^1^ L—low intensity, LC—low-intensity controls, M—medium intensity, MC—medium-intensity controls, H—high intensity, HC—high-intensity controls, ^2^ N—number of individuals trapped, S—number of species, TD—trapping effort, trap days, D—dominance index, H—Shannon’s index of diversity, RA_s_—relative abundance in summer, RA_a_—in autumn. H differences: L–M (*t* = 1.81, *p* = 0.07), L–H (*t* = 1.94, *p* = 0.053), M–H (*t* = 3.21, *p* = 0.002), L–LC (*t* = 1.00, NS), M–MC (*t* = 1.11, NS), H–HC (*t* = −2.56, *p* = 0.01). RA differences in summer (ANOVA, F_2,13_ = 2.65) and autumn (F_2,13_ = 1.81) were not significant between sites with different intensities of agricultural practices.

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
