# Peer review of "Mow the Grass at the Mouse’s Peril: Diversity of Small Mammals in Commercial Fruit Farms"

_animals, 2019, doi:10.3390/ani9060334_

Round 1
Reviewer 1 Report
Not many studies on small mammal community from orchards are from any European country. This study may somehow fill a gap on this problem. Moreover not many information on small mammals are from Baltic countries.
Your evaluation of small mammal communities is strongly based in relation of the agricultural practice intensity and that is why you have to set exactly what the limits are for.
The intensity of agricultural practice was evaluated as follows: high intensity (frequent application of more than two even the same above listed measures, including rodenticides), medium intensity (two listed measures during the crop season even the same) and non-intensive (removal of grass only once?).
In particular sides description you need not repeat agricultural practice intensity as it is in the table 1. But in table 1 the repetition of the measures is not stated.
Relative abundance is always the same definition so don’t repeat again if mentioned (rows 177, 241, 309,312, 315).
Grass mulching is a very favourable for voles enabling to hide against predators and move and also eat if the vegetation already grow again.
Page 328 – Not exactly 40% of sides with rodenticides.
Better organize your tables. First column is better if not centred,
Most of the results is possible to put into summary tables for better overview.
Table A1 is a very informative for to have a good idea about the small mammal communities in the Lithuania country. One thing is exceptional as A. uralensis is in Central Europe inhabiting open land biotopes being dominant species in corn fields and grassy, weedy biotopes. In Lithuania it also goes to young forests and regrowing clearcuts.
Author Response
Comment: The intensity of agricultural practice was evaluated as follows: high intensity (frequent
application
of more than two even the same above listed measures, including
rodenticides), mediumintensity (two listed measures during the crop
season even the same) and non-intensive (removal of grass only once?).
In particular sides description you need not repeat agricultural
practice intensity as it is in the table 1. But in table 1 the
repetition of the measures is not stated.
Answer: After this comment we agree, that it is possible to make text clear – and we made necessary changes. We add description of the frequency of measures used.
Comment: Relative abundance is always the same definition so don’t repeat again if mentioned (rows 177,241, 309,312, 315).
Answer: corrected, repetitions of the text removed.
Comment: Grass mulching is a very favourable for voles enabling to hide against predators and move and also eat if the vegetation already grow again.
Answer: we agree with comment, that mulching at least partially neutralizes effect of the grass mowing, thus, this part of text was not removed in revision.
Comment: Page 328 – Not exactly 40% of sides with rodenticides.
Answer: Yes, we agree, but in this case percentage was rounded on intention. We add “approximately” in revision.
Comment: Better organize your tables. First column is better if not centred,
Rebuttal: template show centered columns, as well as published papers. We consulted last papers from Animals journal, and keep these as acceptable example. We expect, that this may be changed in the layout if needed.
Comment: Most of the results is possible to put into summary tables for better overview.
Answer:
we changed text in the Result part, mainly transferring statistical
part to the table or figure footnotes, and shortened text as requested.
Comment: Table A1 is a very informative for to have a good idea about the small mammal communities in the Lithuania country. One thing is exceptional as A. uralensis is in Central Europe inhabiting open land biotopes being dominant species in corn fields and grassy, weedy biotopes. In Lithuania it also goes to young forests and regrowing clearcuts.
Answer: we agree, and this is seen from Table A1 – however, species was not trapped in the 15 locations, so there is hardly a chance to add text to the manuscript.
Reviewer 2 Report
It is an important work on small mammals in some agricultural ecosystems. Although SM are pests of many crops and are a reservoir for pathogens, they have important ecosystem roles. Many factors influence the wealth and diversity of SM. This article will add scientific data that will be useful for scientists, farmers, managers, I have reviewed the entire article; it is ready to be published in its form. The first author, known for the quality of his publications (Scopus, PubMed, ..), sent an article very important and well prepared. I have a small remark to improve the article. Although the article is ready to be published, the figures may be better in the published version. Figure 1 : Please, it is better to put a clearer map of the country showing the capital, … and of course without showing the location coordinates. Figure 2 and Figure 3: the letter (a) which indicate the figure at the top and (b) which indicate the figure at the bottom are not put in the two figures. In addition, the quality of the figures is lost, the insertion of the figures in the form of images, they lose the quality. It is better to insert them directly from the used software. It is very useful to put the figures S1, S2, ..S5 as appendices at the end of the article or to include them in the article because they are necessary for the reader to better understand the results.
Author Response
Comment: I have a small remark to improve the article. Although the article is ready to be published, the figures may be better in the published version. Figure 1 : Please, it is better to put a clearer map of the country showing the capital, … and of course without showing the location coordinates. Figure 2 and Figure 3: the letter (a) which indicate the figure at the top and (b) which indicate the figure at the bottom are not put in the two figures. In addition, the quality of the figures is lost, the insertion of the figures in the form of images, they lose the quality. It is better to insert them directly from the used software. It is very useful to put the figures S1, S2, ..S5 as appendices at the end of the article or to include them in the article because they are necessary for the reader to better understand the results.
Answer: figures were used for review, now we attach much better quality images. Figure 1 – we attach tif image with 600 dpi resolution, capital shown in colour. Figures 2 and 3 – quality increased, lettering added as required. It is not possible to insert directly images from PAST software, but this program allows change image size, and we used this possibility to increase dpi value. Figures S1-S5 are put into Appendice A, as recommended.
Reviewer 3 Report
This is a potentially interesting study but it is difficult to assess the quality in its current state. I think that the data are not clearly enough presented to convincingly back the conclusions. Most importantly, it is necessary to correct trap days for sprung traps (see detailed comments) or show that sprung traps did not bias the results. For example, if a particular area has more rain than others, the chance that traps are sprung by rain is higher in this area, so abundance estimates without corrections will be biased to a lower value. If you kept data about number of sprung traps, you can easily correct the results. If not, you need to present some kind of evidence that there is no bias. It was also strange to me to find a sort of meta-analysis done in the discussion (data from appendix A) but nothing in the results. If you did a meta-analysis, this needs to be presented in both methods and results.
The introduction is too much written like a discussion and literature review. It needs to be edited towards presenting the necessary information to understand your objectives, not discussing results of previous work. You also talk about hypotheses in three different places, if appropriate, you can write them once at the end, but I do not see any real hypotheses here. They rather seem to be post-hoc hypotheses which can be replaced by clear objectives. In the text of the results, there are by far too many details. You should put the results in tables and figures and use the text to pinpoint the most important results. I cannot find a clear comparison of orchards and crop fields in the results; it may have been in the text but I did not notice this. Or are the crop field results cited from other publications? The discussion mentions a lot of details but does not really help the reader to understand the implications of your study. Why have different types of orchards different small mammal compositions? Is the agricultural method used different or does it also depend on the fruits? You need to point out the most important findings. To me the conclusion is rather a repetition of results but not a conclusion. I strongly recommend having the manuscript improved by a native speaker. Besides grammatical errors (missing articles), the syntax is often unusual. I also recommend avoiding the passive voice.
Detailed comments:
Line 18 (and later). I would not use an abbreviation for small mammals, it makes the manuscript less nice to read. Besides, SM is not usually used for small mammals. “while” and “at the same time” means the same, no need to use both.
Line 19. Use articles (valid for the whole manuscript). Instead of “Common vole is reference species” it should be “the common vole is a reference species”.
Line 23-24. The hypotheses do not make sense to me: “(i) small mammal diversity in fruit farms is low” compared to what? The second hypothesis needs also to be reworded, because it makes no sense that a species is dominant when other species are present, it would mean that it is not dominant when there is no other species. Also, which theoretical reason led you to assume that common voles are always dominant, it is not rather just your observation?
Line 110. You mention here the second time that this is the first study on small mammals in orchard in Baltic countries, in my opinion it was already not necessary the first time.
Line 112. Why 16 trapping sites in 15 study sites? Why not 15 trapping sites?
Line 116 (and elsewhere). If you use the passive voice, you need to specify the actor. Here, for example it is not clear who used rodenticides (farmers or you as experimental setting). Avoid this by using the active voice.
Line 118. Why would the wide distribution of sites simulate temporal changes? Does not convince me.
Fig.1. The map is unreadable. You need to modify it according to the scale and provide only essential information.
Line 129-161. This info should go into table 1, no sense to write all this in the text. I also have the impression it is only a repetition of table 1. Additional info for the photos should go to the figure legends.
Line 167. Which standard method? I think you mean that you used your own standard and published this in a book. However, this book is not available to the English-speaking reader, therefore you need to provide specifics here.
Line 169-171. You mention that trapping did not have an effect on later trapping results but you do not write how often you trapped at each site; you need to specify this first.
Line 172. You write that you replaced bait that was missing but I assume that you did not correct results for unavailable traps. This can significantly bias results if for example one region has more rain (and more traps are closed by it) than other regions. I recommend to correct for sprung traps (see: Nelson, L.Jr. & F.W. Clark. 1973. Correction for sprung traps in catch/effort calculations of trapping results. Journal of Mammalogy 54: 295-298; Beauvais G. P. & Buskirk, S. 1999. Modifying estimates of sampling effort to account for sprung traps. Wildlife Society Bulletin 27: 39-43; Theuerkauf J., Rouys S., Jourdan H., Gula R. 2011. Efficiency of a new reverse-bait trigger snap trap for invasive rats and a new standardised abundance index. Annales Zoologici Fennici 48: 308-318). You need to specify which trap type, size and brand you used. As the small mammals are native species, you should also justify why you used kill traps instead of live traps. Besides the possible effect on later trappings, you should address ethical considerations. You write parts of this later in line 184-186 but it is better to address this matter when you first mention kill traps.
Table 2. It makes more sense to me to integrate sample sizes in result tables (in the result section). You need to specify that you mean trap days when using the term trapping effort (in my opinion this needs to be corrected for sprung/unavailable traps). N is not a good abbreviation for number of individuals as these are not the sample sizes. I do not see a reason to average summer and autumn if densities are not comparable.
Line 188-190. Provide citations for each index you used, otherwise you need to specify how calculated.
Line 195. Not clear what you mean here by rarefaction. Please write exactly what you did.
Line 197. Confidence intervals have no p value. You should write 95% confidence interval. In general, there is no need to mention the tests in this paragraph, you should provide info which test you used every time you report the results.
Line 205-208. Better in a table and with numbers for each species, no need to repeat in the text.
Line 209. You probably mean “observed/recorded species”
Line 213. Test results are of little value if not accompanied by effect size. It is not interesting that there are differences, but how they differ.
Line 209-221. Too many details in the text. Better provide results in a table and then mention the most important results in the text referring to the table.
Line 214 (and elsewhere). Are averages provided with confidence intervals? Please write this the first time when used or write in the methods that all averages are with confidence intervals (if not CI, then do not use plus/minus but write e.g. SE= or SD=. If SD, you need to add also n). Provide the confidence interval with the same precision than the average.
Line 216. Why are these results in the supplements and not in the main article? I also do not see much information stemming from the rarefaction analysis. Is it worth keeping? You might want to delete this whole part from the manuscript for more clarity.
Line 233-237. You need to reword for more clarity.
Line 236-237. This is just an observation, no need to go into hypotheses for this. If you have reason to believe that in general blueberry plantations are no habitats for small mammals, you can address this in the discussion.
Table 3. Numbers trapped is not diversity. If you write diversity in the table title, you need to provide it. Providing raw numbers is good, but it would be better if you do so for each study site and put the raw data table better in the supplement. Mixing raw data and analyses is not a good idea in my opinion. Also explain which kind of variation you are providing. It looks like range in some cases and SE in others (SE is not an interval, so do not write plus/minus. Better use CI here), can’t you provide e.g. CI for all? Do not use N for numbers, better use “nb”. You probably mean “expressed as number of individuals/100 trap days” writing “expressed as standard capture rates to number of animals/100 trap days”. It seems strange to me to use one control group but many orchard types. Should not every type have its own control? If you use CI for each number, then you can directly compare, this is easier and more elegant than significance tests.
Fig 2b. Unless you used exactly the same number of trap days, you should use here abundance indices to make results comparable. I also think that each orchard type needs to be compared with its control. Only if all four controls are equally high, you can show that particular practices reduce small mammal abundance. Especially as you write in line 262 that abundance varied in control habitats from 0 to 12.
Fig.3. The b chart is interesting, but lacks the control habitats. You also need to change to abundance to make the results comparable. How do you explain the high abundance in intensively managed orchards? This is not consistent with your conclusion that intensity of agriculture leads to lower small mammal abundance.
Table 4. Same comments as for table 3. Better provide here only analyses and provide raw data in a supplement (for each site).
Line 327. I have no idea what you want to say in this sentence, please reword.
Line 359. It is difficult to understand what 25-58200 trap days represent? Also 2-3820 individuals. Does this mean there were sites with 25 trap days and 2 individuals and sites with 58200 trap days and 3820 individuals? Please clarify.
Appendix A. I do not fully understand what you mean by data source. Are these the results of your study or is your study a metaanalysis of published data? Reading the paper, I had the impression that you took new data at the 15 sites. Please clarify.
Author Response
Dear Reviewer,
Please find answer to your comments attached
sincerely
Linas Balčiauskas

Round 2
Reviewer 3 Report
The revised manuscript has much improved. However, you have still not considered a major concern. You compare the control sites all combined together against the different orchards. In my opinion, it makes no sense in combining the control habitats because you lose information. This method does not allow assessing if the orchard type actually caused the small mammal abundance/diversity. It is also possible that the differences are caused by regional effects. Therefore, when you present results for each orchard type, you also need to present the results for the associate control sites. The rarefaction curves also do not indicate differences in orchard types, it only indicates that species numbers increase with the number of trapped individuals. As these curves mostly overlap, the difference in species numbers could be purely caused by the number of individuals trapped. Therefore, you need to demonstrate that the differences are not caused by different trapping effort (from table 3 it seems that the diversity was positively correlated to trapping effort). It is also still unclear if other factors than small mammals influenced the trapping results, there is only a rather obscure general comment in the methods. I therefore think that the analyses need changes before the results back up the conclusions.
Some detailed comments:
Line 71-72. I do not think that this hypothesis is valid: “Following [13], we hypothesize that small mammals are able to colonize agricultural habitats when the carrying capacity of the primary habitats is exceeded.” I am not aware of any species that uses exclusively a primary habitat to the carrying capacity and then starts colonising other areas, if the species can live somewhere else, it will.
Lines 92-94. Correct sentence to: “Based on literature sources, we predicted that (i) there would be a low small mammal diversity in commercial gardens and berry fields compared to other habitats and (ii) common vole should be the dominant species.
Line 95. Not clear to which parameters you refer
Fig 1. The map is still full of unreadable details, why don’t you extract only the information you need from the map? You also need to define everything (e.g. the green area, even if this probably is forest)
Line 132. Better provide the exact size than writing medium.
Line 138. “thus a negative influence of trapping was not possible” is incorrect in this context. You need to write “thus the first trapping could not have negatively influenced the second trapping session”. However, this is obvious and does not prove that there was no influence at other sites.
Line 143. Rain is not the only factor biasing trap results, so the absence of rain (only during the first day) does not mean that you do not need to correct your data. The fact that you did not capture any mammal in sites 13 and 15 could have been caused by other animals that sprung the traps. Correction would definitely improve the results. If you cannot correct the trapping because you did not write the number of sprung traps per trapping day, then you might need to exclude the sites with many traps sprung by other factors than small mammals.
Line 164-168. There is still unclarity about rarefaction here. You should use the term rarefaction curves here, so that people know what you mean. However, rarefaction curves do not correct for any influence. You only can see how the number of species evolve by adding more individuals (assuming that the sampling is random). This becomes also evident in the figures showing the rarefaction curves, the curves all start more or less similar, only the number of individuals is different, so these curves do not help you proving that diversity is higher in any habitat as the lower diversity can be everywhere simply caused by lower sample sizes. This reinforces my opinion that you should standardise your data by trapping effort. For example, in fig 2, you still use numbers instead of abundance, which makes the results not comparable.
Author Response
Answers to Reviewer comments
Comment: The revised manuscript has much improved.
1. However, you have still not considered a major concern. You compare the control sites all combined together against the different orchards. In my opinion, it makes no sense in combining the control habitats because you lose information. This method does not allow assessing if the orchard type actually caused the small mammal abundance/diversity. It is also possible that the differences are caused by regional effects. Therefore, when you present results for each orchard type, you also need to present the results for the associate control sites.
2. The rarefaction curves also do not indicate differences in orchard types, it only indicates that species numbers increase with the number of trapped individuals. As these curves mostly overlap, the difference in species numbers could be purely caused by the number of individuals trapped. Therefore, you need to demonstrate that the differences are not caused by different trapping effort (from table 3 it seems that the diversity was positively correlated to trapping effort). It is also still unclear if other factors than small mammals influenced the trapping results, there is only a rather obscure general comment in the methods. I therefore think that the analyses need changes before the results back up the conclusions.
Answer:
1. We acknowledge comment on the separating controls and comparing these to the respective sites (with different crops or different intensity of agricultural practices). Figures 2b and 3b we re-worked to answer comment, including separate controls; data added also to Table 3 and Table 4, as well as additional statistics.
2. We, of course, are not inventers of the rarefaction, thus, in this discussion we refer to published literature sources. For example, Gotelli and Colwell, 2011, page 39: “Although a complete review of the subject is beyond the scope of this chapter, we highlight sampling models for species richness that account for undersampling bias by adjusting or controlling for differences in the number of individuals and the number of samples collected (rarefaction) as well as models that use abundance or incidence distributions to estimate the number of undetected species (estimators of asymptotic richness)” and other sources cited in the manuscript [46,49,50,51]. Thus, we remain on the said differences between small mammal diversity in the different crops, evaluated by PAST software (Table 3, footnote – “H differences: A-P (t = 5.66, p < 0.001), A-C (t = 7.24, p < 0.001), A-R (t = 2.91, p < 0.01), P-C and P-R (NS), C-R (t = 2.40, p < 0.05)“), as species accumulation curves confirm these differences (Figure 2a). The same answer concerns differences between small mammal diversity in the areas with different intensity of agricultural practices (Table 4 and Figure 3a).
In this case, rarefaction is used as supplement to PAST statistics (giving diversity differences between crops being different with various level of significance. However, differences in number of species are big enough even in absolute values: 9 species in apple orchards versus 3 species in plum orchards, 5 species in currant plantations and 6 species in raspberry plantations – this is 1.5-3-fold differences between habitats.
Some detailed comments:
Comment: Line 71-72. I do not think that this hypothesis is valid: “Following [13], we hypothesize that small mammals are able to colonize agricultural habitats when the carrying capacity of the primary habitats is exceeded.” I am not aware of any species that uses exclusively a primary habitat to the carrying capacity and then starts colonising other areas, if the species can live somewhere else, it will.
Answer: In fact, we hypothesize, as original article says „Common voles are a component of agroecosystems in many parts of Europe, inhabiting agricultural areas (secondary habitats) when the carrying capacity of primary grassland habitats is exceeded.“(see below) – we only extend this for other small mammals as hypothesis. However, to avoid further discussion, we exclude last part of the sentence under discussion, and use word „presume“ instead of „hypothesize“.
Comment: Lines 92-94. Correct sentence to: “Based on literature sources, we predicted that (i) there would be a low small mammal diversity in commercial gardens and berry fields compared to other habitats and (ii) common vole should be the dominant species.
Answer: changed as advised.
Comment: Line 95. Not clear to which parameters you refer
Answer: We add text „species composition, abundance and diversity of small mammal communities“ instead of „these parameters“, however, then we repeat text from Line 90-91.
Comment: Fig 1. The map is still full of unreadable details, why don’t you extract only the information you need from the map? You also need to define everything (e.g. the green area, even if this probably is forest)
Answer: According the comment, we removed background of the map and most of administrative information. Added new information show crop type, indicated by different colours.
Comment: Line 132. Better provide the exact size than writing medium.
Answer: corrected as advised, size is 7*14 cm
Comment: Line 138. “thus a negative influence of trapping was not possible” is incorrect in this context. You need to write “thus the first trapping could not have negatively influenced the second trapping session”. However, this is obvious and does not prove that there was no influence at other sites.
Answer: however, emphasis was on the fact, that “influence of the short trap lines on diversity and abundance of small mammals is not found even after long-term trapping up to three times per year”; nevertheless, text was corrected as advised.
Comment: Line 143. Rain is not the only factor biasing trap results, so the absence of rain (only during the first day) does not mean that you do not need to correct your data. The fact that you did not capture any mammal in sites 13 and 15 could have been caused by other animals that sprung the traps. Correction would definitely improve the results. If you cannot correct the trapping because you did not write the number of sprung traps per trapping day, then you might need to exclude the sites with many traps sprung by other factors than small mammals.
Answer: It seems, that sentence “At sites 13 and 15, no small mammals were trapped, hence abundance results were not corrected despite some traps being sprung by slugs, birds or other animals“ should be removed, because is wrongly understood. In the sites 13 and 15 we did not catch any small mammals. There were two trapping sessions – summer and autumn, no rain or sprung traps in summer, all sprung traps rebaited in autumn. And still, no catch. There are two issues related to correction: (1) we defend our point of view to use number of trapped individuals in diversity estimation, thus, we cannot correct these numbers, and (2) abundance estimation in sites 13 and 15 cannot be corrected, as value is zero. In other sites, number of sprung traps is negligible. Yes, we did not write sprung trap numbers, as there were 1, 2, 3 sprung traps per trap line at most. Such correction would not change situation and conclusions.
Comment: Line 164-168. There is still unclarity about rarefaction here. You should use the term rarefaction curves here, so that people know what you mean. However, rarefaction curves do not correct for any influence. You only can see how the number of species evolve by adding more individuals (assuming that the sampling is random). This becomes also evident in the figures showing the rarefaction curves, the curves all start more or less similar, only the number of individuals is different, so these curves do not help you proving that diversity is higher in any habitat as the lower diversity can be everywhere simply caused by lower sample sizes. This reinforces my opinion that you should standardise your data by trapping effort. For example, in fig 2, you still use numbers instead of abundance, which makes the results not comparable.
Answer: first of all term “rarefaction accumulation curves” already was used throughout, however, after checking Gotelli, N.J. and Colwell, R.K., 2011. Estimating species richness. Biological diversity: frontiers in measurement and assessment, 12, pp.39-54., and other cited sources, we changed this to more appropriate “species accumulation curves”.
We do not say “rarefaction curves correct for any influence” as it is said in the comment. In the lines 166-168 it is said “Rarefaction eliminates the influence of the sample size on the diversity estimate” – and we are sure, this is eliminating influence of the trapping effort, as adjusting to various sample sizes IS the very essence of the rarefaction. See, for example, Gotelli and Colwell, 2011, page 39: “Although a complete review of the subject is beyond the scope of this chapter, we highlight sampling models for species richness that account for undersampling bias by adjusting or controlling for differences in the number of individuals and the number of samples collected (rarefaction) as well as models that use abundance or incidence distributions to estimate the number of undetected species (estimators of asymptotic richness)” and other sources cited in the manuscript [46,49,50,51].
Figures 2b and 3b we re-worked to answer comment, including controls; data added also to Table 3 and Table 4, as well as additional statistics.
Secondly, we discussed possibility to base all analysis on data, standardized by trapping effort, and find many examples of published data, that are based on the number of trapped individuals, not relative densities.
From the very beginning, diversity and dominance indices were based on ni, N and pi, where ni is number of individuals of the ith species, N – is sum of individuals of all species, and pi=ni/N. for example:
Peet, R.K., 1974. The measurement of species diversity. Annual review of ecology and systematics, 5(1), pp.285-307.
Whittaker, R. H. (1972). Evolution and Measurement of Species Diversity. Taxon, 21(2/3), 213.doi:10.2307/121819
Somerfield, P.J., Clarke, K.R. and Warwick, R.M. (2008) Simpson index. In: Jorgensen, S.V. and Fath, B., (eds.) Encyclopedia of Ecology. Elsevier, Oxford, UK, pp. 3252-3255. http://dx.doi.org/10.1016/B978-008045405-4.00133-6
So, as for the dominance, “Simpson’s index may be defined in different ways, but the original and simplest is that it is the probability that two individuals drawn at random from an assemblage will belong to the same species. As such it is a measure of dominance, and for a highly dominated (i.e., highly uneven) assemblage the probability of drawing two individuals from the same species will be high (approaching 1). For a completely even assemblage, in which all individuals belong to different species, the probability of drawing two individuals from the same species will be 0. Conventionally, more even assemblages are considered to be more diverse; therefore, this scaling appears counterintuitive as high values imply low diversity. The index is often, therefore, converted from a dominance measure into an evenness (or equitability) measure either by subtracting the dominance value from 1, or by taking its inverse. In comparison to other measures of richness and evenness, Simpson’s index can be shown to be relatively sample-size independent. Its simple definition also suggests methods for its estimation which do not require detailed taxonomic expertise”.
A few papers, using number of individuals, not their standardized value:
Heisler, L.M., Somers, C.M. and Poulin, R.G., 2016. Owl pellets: a more effective alternative to conventional trapping for broad‐scale studies of small mammal communities. Methods in Ecology and Evolution, 7(1), pp.96-103.
Janova, E., & Heroldova, M. (2016). Response of small mammals to variable agricultural landscapes in Central Europe. Mammalian Biology - Zeitschrift Für Säugetierkunde, 81(5), 488–493. doi:10.1016/j.mambio.2016.06.004
Pupila, A. and Bergmanis, U., 2006. Species diversity, abundance and dynamics of small mammals in the Eastern Latvia. Acta univ. latv, 710, pp.93-101.
Ilyashenko, V. B., Luchnikova, E. M., Skalon, N. S., Grebentschikov, I. S., & Kovalevsky, A. V. (2019). Long-Term Dynamics of Small-Mammal Communities in Anthropogenically Disturbed Territories in the South-East of West Siberia. IOP Conference Series: Earth and Environmental Science, 224, 012055.doi:10.1088/1755-1315/224/1/012055
Heroldová, M., Bryja, J., Zejda, J., & Tkadlec, E. (2007). Structure and diversity of small mammal communities in agriculture landscape. Agriculture, Ecosystems & Environment, 120(2-4), 206–210.doi:10.1016/j.agee.2006.09.007
Łopucki, R., & Mróz, I. (2016). An assessment of non-volant terrestrial vertebrates response to wind farms—a study of small mammals. Environmental Monitoring and Assessment, 188(2).doi:10.1007/s10661-016-5095-8 :
Also, switching to data standardised by trapping effort, we will fail to compare data with results of the previous investigations (Table B1). Therefore, we added more standardized results to the Table 3 and Table 4, but remain on the number of individuals.

Round 3
Reviewer 3 Report
The revised manuscript is now transparent so that the reader can directly see your results instead of relying solely on test results. The abundance indices would be better with confidence intervals than with SE so the reader can see where there are differences.